# Anti-Obesity Potential of Barley Sprouts in Dog Diets and Their Impact on the Gut Microbiota

**DOI:** 10.3390/microorganisms13030594

**Published:** 2025-03-04

**Authors:** Hyun-Woo Cho, Kangmin Seo, Min Young Lee, Sang-Yeob Lee, Kyoung-Min So, Seung-Yeob Song, Woo-Duck Seo, Ju Lan Chun, Ki Hyun Kim

**Affiliations:** 1Animal Welfare Research Team, National Institute of Animal Science, Rural Development Administration, Wanju 55365, Republic of Korea; jhwoo3856@korea.kr (H.-W.C.); mylee1231@korea.kr (M.Y.L.); sangnext@korea.kr (S.-Y.L.); ls2237@korea.kr (K.-M.S.); 2Ingredient Examination Diversion, National Agricultural Products Quality Management Service, Ministry of Agriculture, Food and Rural Affairs, Gimcheon 39660, Republic of Korea; kmseo@korea.kr; 3Division of Crop Foundation, National Institute of Crop Science, Rural Development Administration, Wanju 55365, Republic of Korea; s2y337@korea.kr (S.-Y.S.); swd2002@korea.kr (W.-D.S.); 4Academic-Industrial Cooperation Organization, Sunchon National University, Suncheon 57922, Republic of Korea

**Keywords:** canine, pet, food, obesity, weight management, nutrient digestibility, microbiome

## Abstract

Barley sprouts, the germinated and grown leaves of barley, contain various bioactive compounds, including policosanol, saponarin, and lutonarin. The ingestion of barley sprouts may benefit canine weight management, potentially owing to the anti-obesity properties of bioactive compounds. However, there is limited evidence on the efficacy and safety of barley sprout supplementation in dogs. Therefore, through this study, we assessed the impact of barley-sprout-supplemented diet on body weight and health markers in healthy adult beagles over a 16-week period. The results showed a 7.2% reduction in body weight in dogs fed the barley sprout diet. Hematology, complete blood cell count, and blood biochemistry analyses confirmed that all parameters remained within normal ranges, with no significant differences observed between the control and experimental groups. Although the levels of IFN-γ, IL-6, and insulin remained stable, leptin, a hormone associated with body fat, significantly decreased. Further analysis of alterations in the gut microbiota following barley sprout supplementation revealed no significant differences between the control and experimental groups with respect to alpha and beta diversity analysis. The shift at the phylum level, with a decrease in *Firmicutes* and an increase in *Bacteroidetes*, resulted in a reduced Firmicutes/Bacteroidetes ratio. Additionally, the abundance of the *Ruminococcus gnavus* group was high in the experimental group. Functional predictions indicated an enhancement in carbohydrate, amino acid, and cofactor and vitamin metabolism. These findings suggest that a barley sprouts diet is safe for dogs and may offer benefits for weight management through favorable alterations in body weight, hormone levels, and gut microbiota composition.

## 1. Introduction

Obesity represents one of the most significant factors contributing to the deterioration of canine welfare [1]. The prevalence of obesity in dogs has been gradually increasing on an annual basis [2,3,4], leading to an increased incidence of joint disorders caused by excess weight in dogs. Furthermore, there is an increased risk of developing complications such as metabolic disorders, osteoarthritis, and cardiovascular diseases because of this trend [5,6,7,8]. Nevertheless, weight reduction from obesity to a normal weight has been associated with a reduction in osteoarthritis-related disease severity and pain [9], an improvement in insulin sensitivity [10], and a normalization of inflammatory levels [11]. It has also been demonstrated to enhance the quality of life for the canine subject [12], as evidenced by an increase in health-related quality of life scores [13], a commonly utilized clinical measure of quality of life. While a reduction in food intake and an increase in activity are the two primary methods for achieving weight loss in dogs, these methods require a high level of commitment from both the dog and owner [14,15,16]. It is common for dog owners to experience emotional challenges when attempting to reduce their dog’s food intake [17,18]. Additionally, increasing the dog’s activity through walks and outdoor play often requires significant willpower, which can present a practical difficulty for owners [19]. Accordingly, when structuring an appropriate diet for a canine, it is essential to provide food that fulfills their essential nutritional requirements and contains ingredients that facilitate weight management [20].

While dogs are classified as carnivores, their domestication has led them to become omnivorous, allowing them to efficiently digest and utilize carbohydrates as a key energy source [21,22]. Numerous studies have demonstrated that carbohydrates are not only digestible for dogs but also play a vital role in their metabolic processes [23,24,25]. Recent commercial dog foods typically contain 30–60% carbohydrates, with rice and barley being widely used as primary sources due to their recognized nutritional benefits [26,27].

Barley is one of the most widely cultivated cereal grains by humans and has a range of physiological health benefits, including antioxidant, anti-obesity, and blood sugar-lowering properties [28,29,30]. Moreover, cereals in the early stages of germination, such as barley sprouts, have been observed to contain elevated concentrations of diverse bioactive compounds that safeguard the developing seedling from external stressors and facilitate its own germination [31,32,33,34]. Barley sprouts are the leaves within 10 cm of the seed of barley that contain useful components such as lutonarin, saponarin, and policosanol, of which policosanol has been demonstrated to induce the activation of AMP-activated protein kinase (AMPK), a key energy regulator [35]. Saponarin has also been demonstrated to impede the accumulation of lipids in both adipocytes and liver cells (HepG2), whereas lutonarin has been shown to inhibit the NF-κB signaling pathway, which plays a pivotal role in inflammatory responses [36,37]. Collectively, the various bioactive substances present in barley sprouts have been proposed to exert preventive effects against hypertension and obesity, as well as to possess potential as a dietary supplement [36,37,38,39]. Nevertheless, no studies have assessed the efficacy of incorporating barley sprouts into canine nutrition. Considering the findings from previous studies, the potential of barley sprouts as a weight control measure for dogs was investigated with the aims of ascertaining the safety of a diet incorporating barley sprouts, evaluating the efficacy of weight control, and profiling the changes in gut microbiota data caused by barley sprouts.

## 2. Materials and Methods

### 2.1. Animals

All animal experiments were approved by the Institutional Animal Care and Use Committee of the National Institute of Animal Science, South Korea (approval number: NIAS-2019-370). The dogs used in this study were observed according to the ethical guidelines for animal protection.

The animals were adult beagles (aged 3.1 ± 0.05 y, *n* = 14) housed individually in independent rooms (180 cm width, 210 cm length) in the same building at the National Institute of Animal Science of the Rural Development Administration. The dogs were maintained in a constant temperature (22 ± 1 °C) and humidity (50 ± 10%) environment and were provided with food and water ad libitum at the same time of day. The control, non-barley sprout diet group (*n* = 7, four males and three females) and the experimental, barley sprout diet group (*n* = 7, four males and three females), were randomized without distinction between sexes, and all beagles were spayed and neutered. The diets were prepared by the authors; the control diet was based on rice powder, and the experimental diet was formulated with 2.8% barley sprouts with an adjusted proportion of rice powder. The composition was as follows (Table 1). The chemical composition and caloric value of the control and experimental diets were identical, and the feeding amounts were individually adjusted based on metabolic energy requirements (132 kcal/day × body weight [BW]^0.75^kg) of each dog. The diets were fed twice daily (at 10:00 and 17:00), and no residual food was left during the experiment. This protocol was conducted for 16 weeks in accordance with the standards set forth by the Association of American Feed Control Officials. Body weight was assessed at consistent intervals, with measurements taken every two weeks. To account for inter-individual variability within the same group, weight measurements were taken at the outset and conclusion of the study, enabling the calculation of the percentage change in weight for each individual.

### 2.2. Digestibility Analysis

Fecal samples were collected over a five-day period using the whole feces collection method to analyze the digestibility of the nutrients present in the diets. The fecal samples were stored at −20 °C until further analysis was conducted. The dog food and fecal samples were dried in a hot-air oven at 75 °C and homogenized for analysis. The chemical composition was analyzed in accordance with the official standard methods of the Association of Official Analytical Chemists [40]. The digestibility analysis of the nutrients was calculated using the following formula based on the values obtained:$$\mathrm{Digestibility}\;(\%)=(\frac{Nutrient\;intake-Fecal\;nutrient\;excretion}{Nutrient\;Intake})\times 100$$

### 2.3. Blood Samples and Analysis

Blood samples were collected from the cephalic vein of the fore limb simultaneously in each group after 16 weeks of feeding the diets. The collected blood was transferred to ethylenediaminetetraacetic acid (EDTA, ref 367861, BD Vacutainer, Franklin Lakes, NJ, USA) treated tubes and vacutainer serum tubes (ref 367812, BD Vacutainer, Franklin Lakes, NJ, USA), respectively. A blood cell count analysis was conducted using the ProCyte Dx system (IDEXX Laboratories, Westbrook, MA, USA), with the blood samples collected in tubes containing EDTA. A blood biochemical analysis was conducted by centrifuging the blood in Vacutainer serum tubes at 1650 × g for 15 min, after which the supernatant was utilized for analysis with a Hitachi 7180 (Hitachi High-Technologies Co., Tokyo, Japan). The remaining serum was then subjected to enzyme-linked immunosorbent assay (ELISA) analysis for the quantification of IFN-γ (ab193684, Abcam, Cambridge, UK), IL-6 (ab193686, Abcam, Cambridge, UK), insulin (MBS03109696, Mybiosource, San Diego, CA, USA), and leptin (MBS705342, Mybiosource, San Diego, CA, USA) proteins to determine the concentration of obesity-associated proteins in the blood. All ELISA assays were performed according to the manufacturer’s manual.

### 2.4. Fecal Sample Collection and Microbiota Analysis

Naturally passed fecal samples were collected on the same day after 16 weeks of feeding the diets, with the time interval between the first and last sample collection not exceeding one hour, and the samples were stored at −80 °C. DNA was extracted from the feces using the NucleoSpin DNA Stool Kit (Macherey-Nagel, Düren, Germany), and the extracted DNA was amplified from the V3–V4 region (341F/805R) within the 16S rRNA using specific primers (forward: 5′-CTA CGG GNG GCW GCA G-3′, reverse: 5′-GAC TAC HVG GGT ATC TAA TCC-3′) and sequenced by Illumina MiSeq (2 × 300 bp, paired-end sequencing). The generated sequences and raw data were imported into the Quantitative Insights into Microbial Ecology Version November 2020 (QIIME2) for processing, and paired-end reads were merged and demultiplexed to filter out reads with a quality score of less than 20 [41]. The demultiplexed reads were then subjected to primer and adapter trimming with the Divisive Amplicon Denoising Algorithm 2 (DADA2), followed by denoising and chimera removal to generate amplicon sequence variants (ASVs) [42]. The reference genome was the SILVA full-length (version SSU138), and taxonomic classification was analyzed using a naïve-bayes classifier that had been pre-trained on primers in the specific V3–V4 regions used for sequencing [43]. Subsequently, the Phylogenetic Investigation of Communities by Reconstruction of Unobserved States (PICRUSt2) analysis was employed to forecast functional alterations resulting from ecological shifts in the gut microbiome [44], with the annotation of classified functions conducted using the Kyoto Encyclopedia of Genes and Genomes (KEGG) database. The data were analyzed using the Statistical Analyses of Metagenomic Profiles (STAMP) and Kruskal–Wallis test, with a *p*-value of less than 0.05 considered statistically significant [45].

### 2.5. Statistical Analysis

The parameter values, except for microbiota analysis, are presented as the mean ± standard error of the mean (SEM), and statistical analyses were performed using R (version 4.2.3). A paired *t*-test was employed for statistical analysis, with a *p*-value of less than 0.05 considered indicative of statistical significance.

## 3. Results

### 3.1. Effects of the Barley Sprout Diet on Body Weight, Digestibility, and Blood Concentrations of Adipose-Associated Hormones

To ascertain the anti-obesity effects of the barley sprouts diet, the experimental diet was formulated with 2.8% barley sprouts by modifying the proportion of rice powder utilized as the carbohydrate source. Each experimental diet was then administered to beagles for a 16 weeks period, with no discernible difference in the chemical composition and caloric value of the diet compared to the control diet without barley sprouts. The findings revealed that the experimental group fed a diet containing barley sprouts exhibited a mean weight loss of −0.9 ± 0.42 kg after the study period compared to their initial weight (Figure 1A, *p* < 0.05). In the control group, the change in body weight from baseline to endpoint was 0.2 ± 0.19 kg, indicating that for the same caloric intake, only the barley sprout diet reduced body weight. Once the change in body weight of each individual in the group was converted to a percentage, considering the variation in body weight between individuals within the group, it was observed that the percentage change in the body weight of the control group increased from the beginning to the end of the experiment, reaching a value of 101.3 ± 1.4%. The percentage change in body weight of the experimental group was 92.8 ± 3.41%, indicating that the 16-week feeding period on the barley sprout diet resulted in a reduction in body weight (Figure 1B, *p* < 0.05). The apparent total tract nutrient digestibility analysis of the respective diets of the control and experimental groups revealed no statistically significant difference in dry matter (DM), crude protein (CP), acid-hydrolyzed fat, nitrogen-free extract (NFE), and dietary fiber (DF) (Table 2). Based on dry matter, CON (group fed a rice-based diet without barley sprouts) consumed an average of 224.2 ± 9.54 g of feed per day, and BS (group fed a rice-based diet with barley sprouts) consumed 225.7 ± 12.42 g of feed per day, with no significant difference in intake. The digestibility of the DM was 91.7 ± 0.33% and 91.2 ± 0.53% in the control and experimental groups, respectively. The digestibility of CP was 81.5 ± 1.03% in the control group and 79.3 ± 0.81% in the experimental group, whereas the digestibility of NFE was 86.2 ± 1.06% in the control group and 88.3 ± 2.06% in the experimental group, with no statistically significant difference in digestibility between the two groups. Moreover, the metabolic energy was 83.7 ± 0.69% in the control group and 82.0 ± 1.01% in the experimental group, indicating that the addition of barley sprouts did not affect the digestibility of nutrients in the dogs. Among the tests utilizing blood samples, complete blood cell counts (CBCs), which serve as an indirect indicator of overall health and clinical indications, demonstrated no significant differences between the control and experimental groups after 16 weeks of feeding the test diet, except for the red blood cell distribution width (RDW). All other analytes, including erythrocytes, leukocytes, and thrombocyte, exhibited values within the reference range (Table 3). Furthermore, the biochemical analyses of the blood samples from the control and experimental groups, conducted in a diagnostic laboratory setting, revealed no statistically significant differences after 16 weeks of dietary intervention. All measured parameters remained within the reference range (Table 4). Further analysis of blood levels of immune-related (IFN-γ, IL-6) and fat-related (insulin, leptin) hormones at 0 weeks, 16 weeks, and between the control and experimental groups revealed that there were no significant differences in IFN-γ, IL-6, and insulin levels between the control and experimental groups, nor were there any differences in these levels according to diet over the 16-week period (Figure 2). Furthermore, no statistically significant difference in the blood levels of leptin was observed in the control group between 0 weeks and 16 weeks (*p* = 0.28). In contrast, the blood levels of leptin were found to be statistically significantly reduced by 16 weeks of feeding the barley sprouts diet (*p* < 0.05).

### 3.2. Analysis of Gut Microbiota Changes in Response to the Barley Sprout Diet

To evaluate the impact of barley sprout consumption on the gut microbiota, the V3–V4 region of 16S rRNA in fecal samples was analyzed, and the alpha diversity analysis showed the following results (Table 5). No differences were observed between the control and experimental groups in terms of alpha diversity of the gut microbiota, as measured by richness (Observed ASVs, Chao1, ACE), diversity (Shannon, Simpson), and evenness metrics. Furthermore, beta-diversity analysis, a method of comparing groups, demonstrated no significant differences between the control and experimental groups regarding both Jaccard distance and Bray–Curtis dissimilarity (Figure 3). In the taxonomic classification analysis, the dominant taxon in both the control and experimental groups at the phylum level was *Firmicutes* (synonym *Bacillota*), followed by *Bacteroidetes* (synonym *Bacteroidota*), *Fusobacteria*, *Proteobacteria*, and *Actinobacteriota*. *Firmicutes* was identified as the most dominant species in the phylum-level gut microbiota of beagle dogs (Figure 4). As expected, after 16 weeks of feeding a diet supplemented with barley sprouts, the relative abundance of *Firmicutes* decreased (*p* < 0.05) and the relative abundance of *Bacteroidetes* increased (*p* < 0.01). This resulted in a decrease in the Firmicutes/Bacteroidetes (F/B) ratio in the experimental group compared to the control group (*p* < 0.05). Furthermore, at the genus level, only the relative abundance of the *Ruminococcus gnavus* group was observed to be increased in the experimental group in comparison to the control group (*p* < 0.05). As a result of forecasting functional alterations resulting from the modified gut microbiota, many functional alterations were predicted. Among these, those pertaining to the metabolism category at level 1 in the KEGG database were the most dominant (Figure 5). Among the various metabolic functions included in the domain of metabolism, the most frequently observed alterations were in the orders of metabolism of cofactors and vitamins, amino acid metabolism, and carbohydrate metabolism. The metabolism of cofactors and vitamins, including riboflavin metabolism, biotin metabolism, folate biosynthesis, and one carbon pool by folate, were all increased in the experimental group compared to the control group. Furthermore, amino acid metabolism, including glycine, serine, and threonine metabolism; arginine and proline metabolism; tryptophan metabolism; and valine, leucine, and isoleucine biosynthesis, all increased, with only tryptophan metabolism decreasing in the experimental group. Furthermore, the experimental group exhibited increased activity of the citrate cycle, C5-branched dibasic acid metabolism, and glyoxylate and dicarboxylate metabolism, all of which are included in carbohydrate metabolism. Conversely, the experimental group demonstrated decreased activity of glycolysis/gluconeogenesis.

## 4. Discussion

There were no disadvantages in the formulation of the diets with and without barley sprouts in terms of the palatability. The palatability of the dog food was found to be comparable to that of commercially available alternatives, with all dogs consuming the dog food without leaving any leftovers during the trial period. Since no feed residue was observed, it can be evaluated that all individuals consumed enough energy and nutrients to meet their individual needs. In addition, in our study, the content of barley sprout was added at a low level of 2.8%, so it is thought that the addition of sprouted barley did not affect the palatability.

As barley sprouts have yet to be incorporated into canine diets, fecal indices were monitored on a weekly basis for the entirety of the 16-week trial period, and no adverse effects were observed (data not presented). In addition, there were no negative effects associated with the barley sprout diet found across all indicators commonly used in diagnostic laboratory medicine, including complete blood cell and biochemical tests, the indirect indicators of overall health. Furthermore, the experimental diet did not result in any alteration to the blood levels of the immune-associated proteins IFN-γ and IL-6. However, the hematology analysis revealed a statistically significant increase in absolute red cell distribution width (RDW) in the barley sprouts diet group compared to the control group, which exhibited values within the reference range. No numerical difference was observed in mean corpuscular volume and mean corpuscular hemoglobin. Considering the biochemical test results and clinical signs, it is challenging to ascertain whether the observed difference in RDW levels is a consequence of barley sprout feeding.

In this study, we demonstrated weight loss in dogs fed a barley sprout diet for 16 weeks. Barley sprouts can facilitate a reduction in body weight in C57BL/6J mice when they are fed a high-fat diet along with barley sprouts [46]. In vitro studies have demonstrated that barley sprouts inhibit the accumulation of fat in both 3T3-L1 and HepG2 cells [36]. The anti-obesity effect of barley sprouts was evaluated to determine its impact on digestibility; however, no difference in digestibility was observed with the addition of barley sprouts in dogs. The digestibility of commercially available dog foods has been reported to range from 66.9% to 84.4% for dry matter (DM), 70.4% to 82.5% for crude protein (CP), 76.1% to 95.8% for crude fat, 65.0% to 87.6% for organic matter, and 72.6% to 87.7% for energy [47,48]. In a previous study, the digestibility of rice-based diets was similar to that of the control diet containing no barley sprouts used in this experiment [20]. This study found that the DM-based digestibility of the barley sprout diets ranged above the 90% threshold, and all other nutrients were well-digested by dogs compared to commercial diets. Following the administration of the barley sprout diet, a reduction in the concentration of the obesity-associated protein leptin was observed. The concentration of leptin present in the bloodstream in a non-fasting state is significantly correlated with the development of obesity because of its expression in adipose tissue [49]. Although further scientific research is required to confirm this hypothesis, existing evidence, including numerous reports of leptin being associated with obesity and the phenotypic characterization of body weight, suggests that reduced blood leptin concentrations indicate weight loss by feeding barley sprouts.

Dietary fiber (DF) is closely associated with changes in the gut microbiota, and among different DFs, barley has been reported to significantly alter the gut microbiota in dogs [50]. The ingestion of barley leaves has been demonstrated to prevent gut dysbiosis and enrich inosine, a metabolite of the gut microbiota. Inosine has been shown to activate PPARγ and enhance gut mucosal barrier function [51]. Nevertheless, research on barley sprouts remains scarce, and there is a dearth of literature examining the impact of barley sprout consumption on the gut microbiota.

The major phyla that comprise the canine gut microbiota are *Firmicutes*, *Bacteroidetes*, *Fusobacteriota*, *Proteobacteria*, and *Actinobacteria*, supporting previous findings [52]. A reduction in the population of *Bacteroidetes* and *Firmicutes* was observed at the phylum level following the feeding of barley sprouts. *Firmicutes* and *Bacteroidetes* are dominant phyla that influence host energy metabolism and immune responses [53,54]. An increase in the F/B ratio, characterized by an increase in the relative abundance of *Firmicutes* and a decrease in the relative abundance of *Bacteroidetes*, has been reported to be induced by high-fat diet consumption [55,56]. Compared to fiber-rich diets, protein-rich diets increase *Firmicutes* and decrease *Bacteroidetes* at the phylum level in the gut microbiota [57]. A plant-based diet composition has been demonstrated to result in an increase in the abundance of *Bacteroidetes* when compared to diets comprising animal fat and protein [50]. Furthermore, *Firmicutes* demonstrate superior efficacy in the extraction of energy from food sources in comparison to *Bacteroidetes* and have been observed to promote great energy production [53]. This reduction in the F/B ratio was induced by functional anti-obesity agents such as garcinol and resveratrol, demonstrating preventive effects against obesity and metabolic syndrome [56,58]. These studies commonly reported an increase in the F/B ratio of the gut microbiota in obese individuals [59]. Moreover, the overall composition of the gut microbiota can be influenced by diet to a greater extent than by changes caused by environmental factors [60]. The aforementioned results may indicate a potential correlation between weight loss and the F/B ratio, particularly in the context of barley sprout consumption. However, the association between an increased F/B ratio and obesity is not absolute, as a balanced ratio is indicative of a healthy gut microbiota environment [59]. In contrast, studies on the relationship between the F/B ratio and obesity in dogs are significantly limited, highlighting the need for further data accumulation.

The observed increase in the *Ruminococcus gnavus* group at the genus level is also attributable to the presence of non-digestible carbohydrates, including resistant starch and whole-grain forms of barley [61]. Nevertheless, as the ideal gut microbiota is more concerned with ecological balance and diversity than the metabolic processes of individual microorganisms, further research is required to gather data on the interaction between the ideal F/B ratio for the gut microbiota of healthy dogs and the abundance of *Ruminococcus*, as influenced by barley sprout consumption. Various analytical methods, such as marker gene analysis, whole metagenome analysis, and metatranscriptome analysis, are available for microbiome research and must be carefully considered to obtain meaningful experimental results. The marker gene analysis used in this study is characterized by a very low sequencing error rate [62] and is most suitable for a wide range of sample types and study designs. It is generally conducted prior to whole metagenome and metatranscriptome analyses to provide a foundational understanding of microbial communities, upon which additional analyses can be considered [63]. In the case of dogs, the accumulation of reference genome databases remains limited. Given the current experimental conditions involving cross-sectional studies, marker gene analysis was selected as the most appropriate method. However, further in-depth studies are essential to achieve species-level resolution and to elucidate the functional roles of actively expressed genes and microbial communities. Nevertheless, it is possible to infer biological functions associated with microbial communities by linking features derived from 16S rRNA gene sequences with marker gene data [44,64,65]. Therefore, as a result of predicting functional changes by changes in the overall microbiota, distinct features of the microbial metabolic profile were observed by barley sprouts consumption. The reduction in weight and decrease in the phosphotransferase system (PTS) observed in barley sprouts were consistent with the findings of other studies that have demonstrated an increase in PTS in obese individuals [66,67]. Furthermore, a high intake of fat and sugar was found to be associated with an increased abundance of PTS [67]. This finding is consistent with the results of a previous study that has observed a reduction in geraniol degradation in individuals with obesity and an increase in geraniol degradation in those with improved obesity [68]. Given that *Bacteroidetes* encode a greater number of carbohydrate-degrading enzymes than *Firmicutes*, increases were identified in the TCA cycle, C5-branched dibasic acid metabolism, glyoxylate, and dicarboxylate metabolism, whereas there were decreases in glycolysis/gluconeogenesis. Furthermore, the biosynthesis of lipopolysaccharides and N-glycans, the glycan metabolisms diminished in obesity, were enhanced [66,69]. The impact of different diets on obesity-associated PTS, geraniol degradation, and carbohydrate metabolism suggests that these processes may be susceptible to dietary modulation.

## 5. Conclusions

In this study, we evaluated the potential of barley sprouts as an anti-obesity functional feed for dogs and expanded the existing information on their association with obesity through their effects on the gut microbiota. Our findings indicate that the consumption of barley sprouts has no negative effects on beagle health and contributes to weight reduction. Furthermore, barley sprouts may promote a shift in the gut microbiota ecosystem toward a non-obese state, supporting their role in weight management. Further investigations are warranted to fully elucidate the mechanisms underlying the beneficial properties of barley sprouts. However, the findings of this study suggest that barley sprouts may be suitable as a feed ingredient for beagles with potential applications in weight control.

## Figures and Tables

**Figure 1 microorganisms-13-00594-f001:**
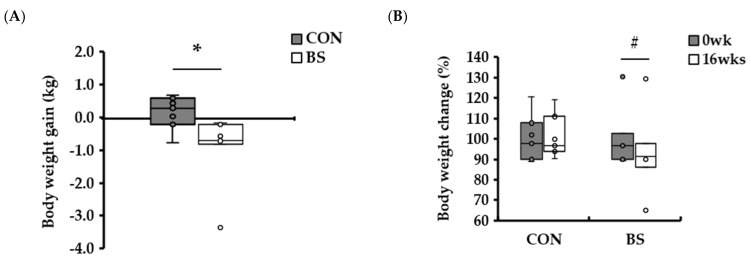
Body weight changes in dogs fed with experimental diets. (**A**) Body weight gain in CON and BS at the initiation and end of the experiment. (**B**) Rate of change of body weight in CON and BS at the initiation and the end of the experiment. * *p* < 0.05 versus CON. # *p* < 0.05 versus 0 weeks. CON, group fed a rice-based diet without barley sprouts; BS, group fed a rice-based diet with barley sprouts. *n* = 7 per group.

**Figure 2 microorganisms-13-00594-f002:**
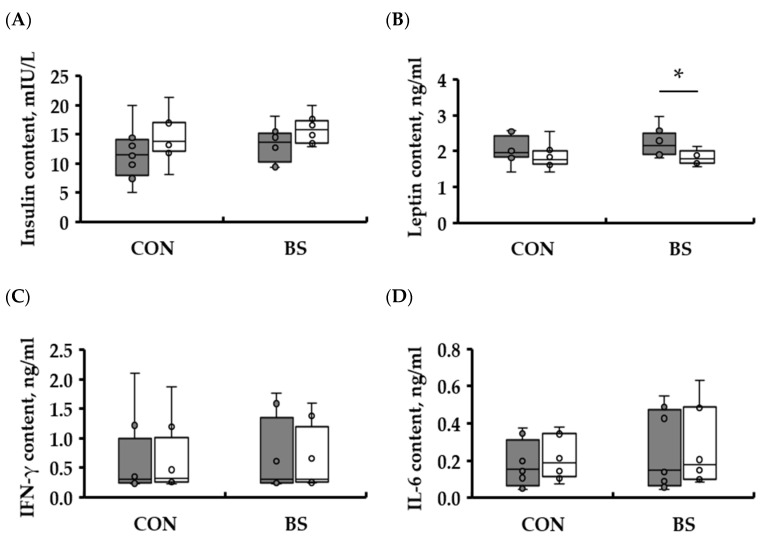
The concentrations of immunoglobulin, cytokines, and hormones in the serum of dogs fed with rice-based diets with or without barley sprouts. (**A**) Insulin, (**B**) leptin, (**C**) IFN-γ, and (**D**) IL-6 contents at the initiation and end of the experiment. Data are expressed as the mean ± SEM. Gray boxes represent before feeding the experimental diet, and white boxes represent after 16 weeks of being fed the experimental diet. CON, group fed a rice-based diet without barley sprouts; BS, group fed a rice-based diet with barley sprouts; * *p* < 0.05 versus 0 weeks. *n* = 7 per group.

**Figure 3 microorganisms-13-00594-f003:**
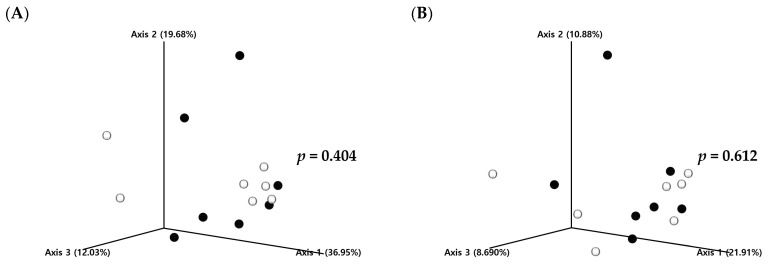
Principal coordinates analysis (PCoA) plots based on amplicon sequence variants (ASVs) identified using QIIME2. (**A**) Bray–Curtis dissimilarity and (**B**) Jaccard distance. Each axis represents the percentage of variance explained in the distance matrix. Bray-Curtis dissimilarity captures differences in the relative abundance of ASVs, whereas Jaccard distance reflects differences based on the presence or absence of ASVs. Black circles represent the group fed a rice-based diet without barley sprouts, whereas white circles represent the group fed a rice-based diet with barley sprouts.

**Figure 4 microorganisms-13-00594-f004:**
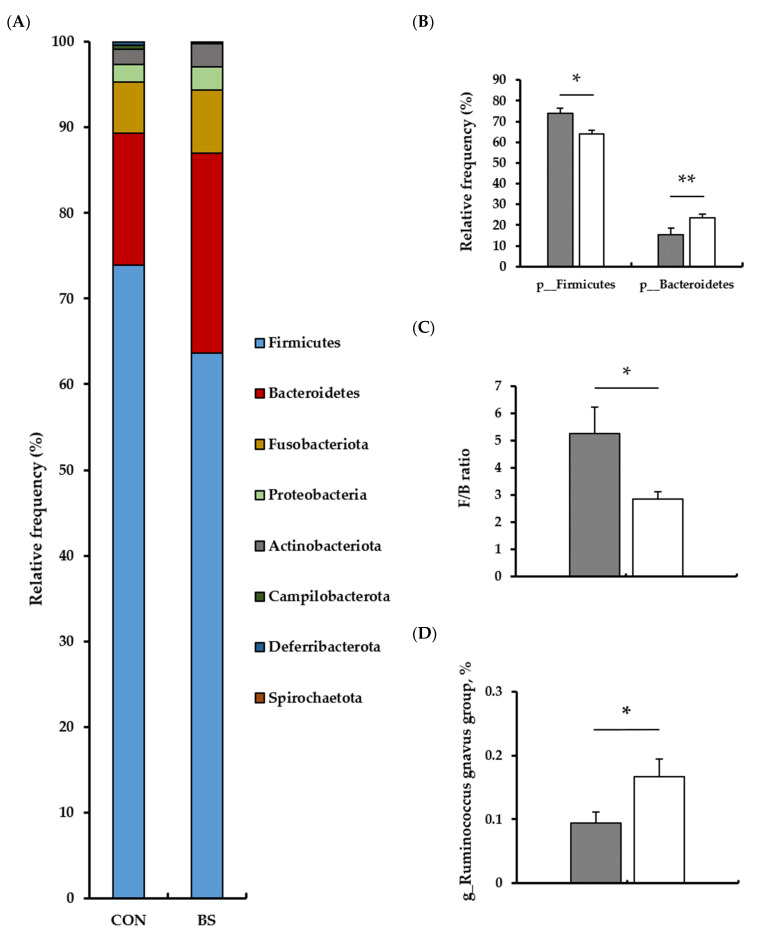
Taxonomic classification of the dog gut microbiome. (**A**) Relative frequency of amplicon sequencing variants at the phylum level for the CON and BS groups. (**B**) Relative abundance of the dominant phyla, *Firmicutes* and *Bacteroidetes*, in both groups. (**C**) Firmicutes-to-Bacteroidetes (F/B) ratio. (**D**) Differences in the relative abundance of *Ruminococcus gnavus* group at the genus level, belonging to the phylum *Firmicutes*, between CON and BS. Gray bars represent CON, and white bars represent BS, both after 16 weeks of experimental diet feeding. CON, group fed a rice-based diet without barley sprouts; BS, group fed a rice-based diet with barley sprouts; p_, phylum; g_, genus; * *p* < 0.05, ** *p* < 0.01 versus CON. *n* = 7 per group.

**Figure 5 microorganisms-13-00594-f005:**
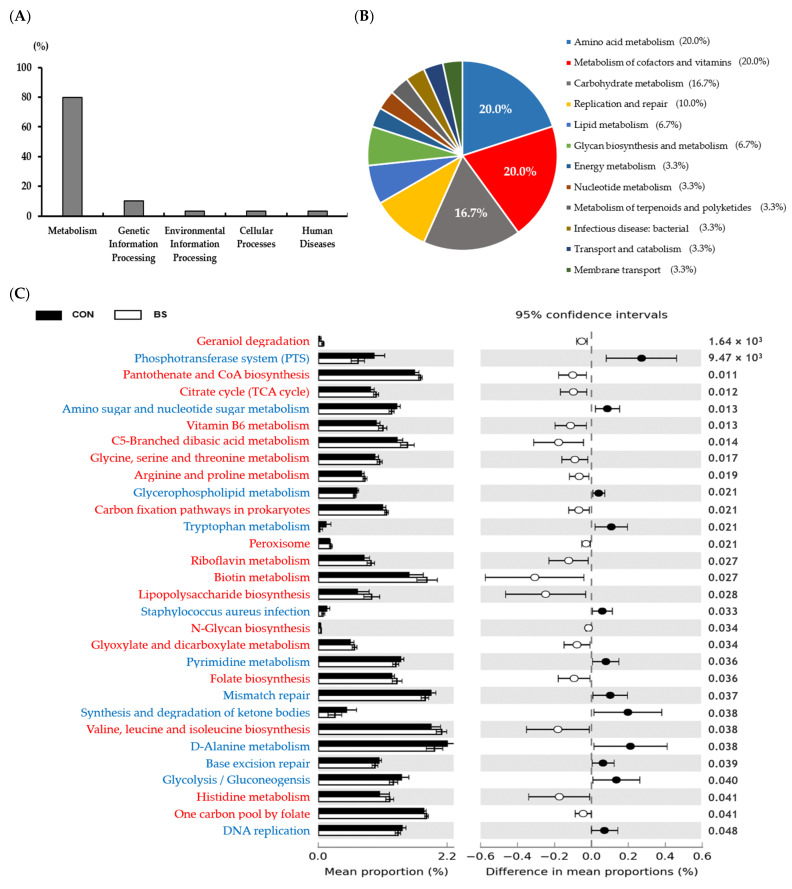
Phylogenetic investigation of communities via the reconstruction of unobserved states (PICRUSt2) analysis results of predicted functional Kyoto encyclopedia of genes and genomes (KEGG) pathway between the rice-based without barley sprouts and rice-based with barley sprouts in fecal microbiota. KEGG metabolism category classifications at level 1 (**A**), level 2 (**B**), and level 3 (**C**). Increased KEGG metabolism categories at the third hierarchy level in BS relative to CON are highlighted in red, whereas decreased pathways are highlighted in blue. CON, group fed a rice-based diet without barley sprouts; BS, group fed a rice-based diet with barley sprouts.

**Table 1 microorganisms-13-00594-t001:** Ingredient formulations and chemical compositions of experimental diets.

Item	CON	BS
Ingredient, %		
Rice powder	31.9	29.2
Barley sprout powder	N/A	2.8
Lard	1.5	1.5
Water	35.0	35.0
Salt	0.2	0.2
Vitamin and mineral premix ^1^	0.4	0.4
Calcium phosphate	0.4	0.4
Potassium citrate	0.6	0.6
Cabbage powder	1.0	1.0
Calcium carbonate	1.0	1.0
Green laver	1.0	1.0
York powder	12.0	12.0
Chicken breast meal	15.0	15.0
Chemical composition, DM basis (analyzed), %		
Crude protein	39.5	38.9
Crude fat	18.1	18.3
Crude ash	8.2	8.4
Crude fiber	2.0	2.8
Nitrogen-free extract	32.2	31.6
Calcium	0.83	0.81
Phosphorus	0.59	0.57
Metabolizable energy, kcal/kg ^2^	4045	4021

^1^ Vitamin and mineral premix supplied per kg of diets: 3500 IU vitamin A; 250 IU vitamin D3; 25 mg vitamin E; 0.052 mg vitamin K; 2.8 mg vitamin B1(thiamine); 2.6 mg vitamin B2 (riboflavin); 2 mg vitamin B6 (pyridoxine); 0.014 mg vitamin B12; 6 mg Cal-d-pantothenate; 30 mg niacin; 0.4 mg folic acid; 0.036 mg biotin; 1000 mg taurine; 44 mg FeSO_4_; 3.8 mg MnSO_4_; 50 mg ZnSO_4_; 7.5 mg CuSO_4_; 0.18 mg Na_2_SeO_3_; 0.9 mg Ca(IO_3_)_2_. ^2^ Metabolizable energy (ME) was calculated following an equation; ME (kcal/kg) = ((CP × 3.5) + (EE × 8.5) + (NFE × 3.5)) × 10. CON, rice-based diet without barley sprout; BS, rice-based diet with barley sprouts; DM, dry matter; CP, crude protein; EE, ether extract; NFE, nitrogen-free extract.

**Table 2 microorganisms-13-00594-t002:** Nutrient intake and apparent total tract nutrient digestibility in dogs fed with rice-based diets with or without barley sprouts.

	CON	BS	*p*-Value
Daily DM intake			
DM (g)	224.2 ± 9.54	225.7 ± 12.42	0.929
CP (g)	78.3 ± 3.33	77.0 ± 4.24	0.814
AHF (g)	25.6 ± 1.09	23.6 ± 1.30	0.284
NFE (g)	110.6 ± 4.71	115.3 ± 6.34	0.569
OM (g)	214.4 ± 9.13	215.9 ± 11.88	0.926
ME (kcal/kg) ^1)^	941.3 ± 40.06	944.6 ± 51.99	0.958
ATTD (%)			
DM	91.7 ± 0.33	91.2 ± 0.53	0.576
CP	81.5 ± 1.03	79.3 ± 0.81	0.253
AHF	95.8 ± 0.25	93.1 ± 1.30	0.170
NFE	86.2 ± 1.06	88.3 ± 2.06	0.534
DF	41.9 ± 2.78	40.6 ± 1.75	0.779
ME	83.7 ± 0.69	82.0 ± 1.01	0.332

Values are expressed as the mean ± SEM. DM, dry matter; CP, crude protein; AHF, acid-hydrolyzed fat; NFE, nitrogen-free extract; OM, organic matter; ME, metabolic energy; DF, dietary fiber; ATTD, apparent total tract nutrient digestibility; CON, group fed a rice-based diet without barley sprouts; BS, group fed a rice-based diet with barley sprouts. ^1)^ ME was calculated using the following equation: ME (kcal/kg) = (CP × 3.5) + (EE × 8.5) + (NFE × 3.5).

**Table 3 microorganisms-13-00594-t003:** Complete blood cell counts in dogs fed with rice-based diets with or without barley sprouts.

Parameter, Unit	Reference Range	CON	BS	*p*-Value
Red blood cell, ×10^6^/μL	5.65–8.87	8.1 ± 0.18	8.4 ± 0.12	0.217
Hematocrit, %	37.3–61.7	52.3 ± 0.93	52.0 ± 1.00	0.854
Hemoglobin, g/dL	13.1–20.5	18.2 ± 0.35	18.2 ± 0.35	0.978
Mean corpuscular volume, fL	61.6–73.5	64.7 ± 0.58	62.2 ± 1.06	0.058
Mean corpuscular hemoglobin, pg	21.2–25.9	22.5 ± 0.22	21.7 ± 0.40	0.107
Mean corpuscular hemoglobin concentration, g/dL	32–37.9	34.7 ± 0.14	34.9 ± 0.16	0.474
Red cell distribution width, %	13.6–21.7	19.6 ± 0.22	20.3 ± 0.25	0.040
Reticulocytes, ×10^3^/μL	10–110	46.8 ± 11.91	42.6 ± 5.77	0.622
White blood cell, ×10^3^/μL	5.1–16.8	9.2 ± 0.34	8.4 ± 0.82	0.366
Neutrophil, ×10^3^/μL	3.0–11.6	6.2 ± 0.38	5.3 ± 0.62	0.228
Lymphocyte, ×10^3^/μL	1.1–5.1	2.4 ± 0.14	2.5 ± 0.21	0.486
Monocyte, ×10^3^/μL	0.2–1.1	0.3 ± 0.03	0.3 ± 0.03	0.651
Eosinophil, ×10^3^/μL	0.1–1.2	0.2 ± 0.02	0.2 ± 0.03	0.297
Basophil, ×10^3^/μL	0–0.1	0.0 ± 0.00	0.0 ± 0.00	0.786
Platelets, ×10^3^/μL	184–484	295.7 ± 28.78	282.6 ± 35.79	0.780

Values are expressed as the mean ± SEM. CON, group fed a rice-based diet without barley sprouts; BS, group fed a rice-based diet with barley sprouts.

**Table 4 microorganisms-13-00594-t004:** Serum biochemistry in dogs fed with rice-based diets with or without barley sprouts.

Parameter, Unit	Reference Range	CON	BS	*p*-Value
Total protein, g/dL	5.4–7.7	6.6 ± 0.11	6.4 ± 0.14	0.247
Aspartate transaminase, U/L	19–42	26.3 ± 2.25	28.0 ± 3.21	0.641
Alanine transaminase, U/L	19–67	32.0 ± 3.38	32.8 ± 5.39	0.902
Gamma-glutamyl transferase, U/L	0–6	4.3 ± 1.12	3.8 ± 0.27	0.650
Creatinine, mg/dL	0.5–1.7	0.8 ± 0.04	0.8 ± 0.03	0.830
Glucose, mg/dL	76–119	103.4 ± 2.46	98.8 ± 2.05	0.144
Lactate dehydrogenase, U/L	0–236	90.5 ± 11.26	109.3 ± 13.61	0.275
Cholesterol, mg/dL	135–361	228.4 ± 15.41	235.9 ± 21.44	0.766
Triglycerides, mg/dL	19–133	61.9 ± 11.45	41.1 ± 7.62	0.129
Urea nitrogen, mg/dL	8–28	13.4 ± 0.96	14.2 ± 0.59	0.446
Total bilirubin, mg/dL	0–0.51	0.1 ± 0.01	0.1 ± 0.01	0.070
Creatine kinase, U/L	52–368	147.0 ± 11.33	176.5 ± 34.73	0.402

Values are expressed as the mean ± SEM. CON, group fed a rice-based diet without barley sprouts; BS, group fed a rice-based diet with barley sprouts.

**Table 5 microorganisms-13-00594-t005:** Analysis of gut microbial alpha diversity in dogs after feeding diets with barley sprout.

Index	CON	BS	*p*-Value
Chao1	151.6 ± 9.60	162.2 ± 16.00	0.581
Shannon	4.9 ± 0.07	5.0 ± 0.25	0.660
Simpson	0.9 ± 0.00	0.9 ± 0.01	0.919
Evenness	0.7 ± 0.00	0.7 ± 0.03	0.742
Ace	152.3 ± 9.65	163.7 ± 16.15	0.556
Observed features	137.1 ± 14.06	161.6 ± 16.06	0.275

Data are expressed as the mean ± SE. Indices of alpha diversity of the gut microbiota, measured by richness (Observed ASVs, Chao1, ACE), diversity (Shannon, Simpson), and evenness metrics. CON, group fed a rice-based diet without barley sprouts; BS, group fed a rice-based diet with barley sprouts.

## Data Availability

The original contributions presented in the study are included in the article; further inquiries can be directed to the corresponding author.

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
