# Peer review of "Anti-Obesity Potential of Barley Sprouts in Dog Diets and Their Impact on the Gut Microbiota"

_microorganisms, 2025, doi:10.3390/microorganisms13030594_

Round 1
Reviewer 1 Report (Previous Reviewer 3)
Comments and Suggestions for Authors
microorganisms-3512205
Authors have partially improved the manuscript, adding some additional statements and corrections. Maybe paper can be accepted for publication, if the Editor of the Journal is satisfied with the corrections and justifications provided by authors. However, taking into consideration general statement regarding feed preferences of the dogs, I think that authors will need to provide references regarding text added into the revised version.
Moreover, when fecal samples were analyzed, authors obtained DNA and used them as a matrix for the investigation. Somewhere in the text, authors need to state limitations on microbiome analysis when DNA and not RNA was used.
L57-62: this statements needs references
Ln146: DNA was isolated and used as material for metagenomic analysis. Please, justify why DNA and not RNA was used and later in the discussion provide comments on the limitation in interpretation of the results, when DNA and not RNA was applied.
Author Response
Authors have partially improved the manuscript, adding some additional statements and corrections. Maybe paper can be accepted for publication, if the Editor of the Journal is satisfied with the corrections and justifications provided by authors. However, taking into consideration general statement regarding feed preferences of the dogs, I think that authors will need to provide references regarding text added into the revised version.
Moreover, when fecal samples were analyzed, authors obtained DNA and used them as a matrix for the investigation. Somewhere in the text, authors need to state limitations on microbiome analysis when DNA and not RNA was used.
L57-62: this statements needs references
Response: Thank you for pointing this out. We have now added appropriate references to support the statements regarding dog feed preferences in the revised manuscript (Lines 57-62). These references provide scientific evidence on dietary choices and feeding behaviors in dogs, ensuring the validity of our experiments.
Ln146: DNA was isolated and used as material for metagenomic analysis. Please, justify why DNA and not RNA was used and later in the discussion provide comments on the limitation in interpretation of the results, when DNA and not RNA was applied.
Response: We have added the relevant content addressing the reviewer's comments in the Discussion section (Lines 390-404)

Reviewer 2 Report (Previous Reviewer 2)
Comments and Suggestions for Authors
Minor comments:
- Line 192: CON should be defined.
- In Figure legend 1: Define CON and BS.
- Table 5: Define the meaning of each of the indices.
- Figure 3: Please define in legend which parameters are used for the three axes.
- Please indicate in Figure 4 the phylum to which Ruminococcus gnavus
Author Response
Line 192: CON should be defined.
Response: We have added the definition of CON in Line 192.
In Figure legend 1: Define CON and BS.
Response: We have revised the legend of Figure 1 to include definitions for CON and BS.
Table 5: Define the meaning of each of the indices.
Response: We have added explanations for each of the indices in Table 5.
Figure 3: Please define in legend which parameters are used for the three axes.
Response: We have revised the legend of Figure 3 to specify the parameters used for the three axes.
Please indicate in Figure 4 the phylum to which Ruminococcus gnavus
Response: We have updated Figure 4 to indicate the phylum classification of Ruminococcus gnavus..

Reviewer 3 Report (Previous Reviewer 1)
Comments and Suggestions for Authors
The article can be recommended for publication.
Author Response
The article can be recommended for publication.
Response: We sincerely appreciate the reviewer’s time and effort in evaluating our manuscript. Thank you for your positive feedback and recommendation for publication.

This manuscript is a resubmission of an earlier submission. The following is a list of the peer review reports and author responses from that submission.
Round 1
Reviewer 1 Report
Comments and Suggestions for Authors
The authors conducted a deep and high-quality experiment on the effect of barley sprouts in the diet of dogs on the prevention of obesity in animals. The data are correctly displayed in tables and figures. The article is written qualitatively, at a high scientific level.
Disadvantages of the manuscript.
1. It is better to remove the words "Assessing the" from the title of the article.
2. The keywords are unsuccessful, they should not duplicate the words of the title. It is better to add new 6-7 key phrases by which the authors would try to find similar studies in the Scopus or Web of Science databases.
3. Line 82: what breed of animals? Line 88: what is the sex of the animals?
4. In Figure 1B, the minimum value should be made 80%, the maximum - 110%, divisions on the ordinate axis - at 2% intervals. Similar changes should be made in Figures 2A and 2B. In Figure 2, the same legend should not be repeated 4 times. The inscriptions above the figures should be removed. On the ordinate axis write "Insulin content comma mIU/L". On lines 209, 234 and 284 you need to indicate the method of comparing samples and the repetition for each variant of the experiment (7-fold repetition everywhere?).
5. The width of tables 3 and 4 allows you to write the full names in the first column, make the table convenient for readers to read.
6. Figure 3 does not allow you to see any patterns: you need to add vertical or horizontal projections of points on the plane.
7. I recommend making Figure 4A 2.5 times higher: this will allow readers to see the values ​​​​for all groups of bacteria. Place Figures 4b, 4C and 4D on the right, move the inscription above Figure 4D to the title of the ordinate axis. There is no need to repeat the legend three times. I recommend drawing a gray column instead of a black column in all figures (Figs. 1, 2, 4, 5). On the ordinate axis of Figure 4D, all numbers should be rounded to hundredths. 8. Table 5: rounding of both the mean and the standard error of the mean should be the same for each row of the table (except for the last column, where the authors correctly rounded all the figures to thousandths).
9. Figure 5B: % should be written next to each of the sectors.
10. Do Figures 5A and 5B show the control or experimental group of animals? Is it possible to make two figures: for the control and experimental groups of animals?
11. Figure 5C needs a more in-depth analysis: if possible, I would recommend that the authors add a large figure with “KEGG metabolism category classifications” to the Discussion, adding formulas of substances to it and indicating in color which of the metabolic blocks is enhanced and which is weakened. This is really interesting for readers. In a simplified and modified form, this figure can become a graphic abstract of the article. Such an addition will significantly increase the number of citations of this article.
12. Line 308-313: What determines the decrease in body weight of animals: (1) cellulose due to a decrease in the caloric content of feed (fewer calories - less weight gain)?, (2) enrichment of the bacterial flora of the intestine due to the addition of food substrates of bacteria - pectins and cellulose - to the food of dogs leads to a decrease in the consumption of calories from food by dogs?, (3) vitamins, amino acids and secondary metabolites of plants - possible analogues of animal hormones (secondary metabolites interact with receptors of dog hormones)?, (4) an increase in the immune response in the intestine to enriched microflora and due to this a switch in the dog's metabolism from the accumulation of fat to the possible fight against potentially pathogenic bacteria? or something else? In my opinion, the fourth answer option is the most correct. It is necessary to study this issue in more detail in the discussion.
Author Response
Response to Reviewers' comments
I appreciate very much for reviewer’s helpful comments on this manuscript. The manuscript has been revised according to reviewer’s suggestion thoroughly and details are explained below.
Reviewer #1:
The authors conducted a deep and high-quality experiment on the effect of barley sprouts in the diet of dogs on the prevention of obesity in animals. The data are correctly displayed in tables and figures. The article is written qualitatively, at a high scientific level.
Disadvantages of the manuscript.
- It is better to remove the words "Assessing the" from the title of the article.
Response: In response to the reviewer comments, the title has been revised by removing "Assessing the," resulting in the updated title: "Anti-Obesity Potential of Barley Sprouts in Dog Diets and Their Impact on Gut Microbiota."
- The keywords are unsuccessful, they should not duplicate the words of the title. It is better to add new 6-7 key phrases by which the authors would try to find similar studies in the Scopus or Web of Science databases.
Response: In response to the reviewer comments, the keywords have been revised as follows: Canine; Pet; Food; Obesity; Weight management; Nutrient digestibility; Microbiome.
- Line 82 and 88: What are the breed and sex of the animals?
Response: Breed, sex, neutralization or not, and grouping information about the dogs used in the study have been added in the Animal section of Materials and Methods.
- In Figure 1B, the minimum value should be made 80%, the maximum - 110%, divisions on the ordinate axis - at 2% intervals. Similar changes should be made in Figures 2A and 2B. In Figure 2, the same legend should not be repeated 4 times. The inscriptions above the figures should be removed. On the ordinate axis write "Insulin content comma mIU/L". On lines 209, 234 and 284 you need to indicate the method of comparing samples and the repetition for each variant of the experiment (7-fold repetition everywhere?).
Response: Thank you for the valuable feedback. Revisions have been made in accordance with the reviewer comments.
- The width of tables 3 and 4 allows you to write the full names in the first column, make the table convenient for readers to read.
Response: In accordance with the reviewer comments, full names have been used to enhance readability for readers.
- Figure 3 does not allow you to see any patterns: you need to add vertical or horizontal projections of points on the plane.
Response: As diet differences can influence the gut microbiota environment, differences in the gut microbiota ecosystem between groups were examined. A diet supplemented with 2.8% sprouted barley fed for 16 weeks had no significant impact on the overall gut microbiota ecosystem in dogs (Table 5, Figure 3). To improve understanding of the content, the positions of the p-values and image quality of figure have been adjusted.
- I recommend making Figure 4A 2.5 times higher: this will allow readers to see the values for all groups of bacteria. Place Figures 4b, 4C and 4D on the right, move the inscription above Figure 4D to the title of the ordinate axis. There is no need to repeat the legend three times. I recommend drawing a gray column instead of a black column in all figures (Figs. 1, 2, 4, 5). On the ordinate axis of Figure 4D, all numbers should be rounded to hundredths.
Response: In accordance with the reviewer comments, Figure 4 has been reorganized, and the black bars in all graphs have been changed to gray.
- Table 5: rounding of both the mean and the standard error of the mean should be the same for each row of the table (except for the last column, where the authors correctly rounded all the figures to thousandths).
Response: The mean values and standard errors in all tables have been standardized to the same number of decimal places.
- Figure 5B: % should be written next to each of the sectors.
Response: In accordance with the reviewer comments, percentages (%) have been added next to each sector.
- Do Figures 5A and 5B show the control or experimental group of animals? Is it possible to make two figures: for the control and experimental groups of animals?
Response: Figure 5 is a representation of the KEGG metabolism categories that show differences between the control and experimental groups, organized by hierarchy level.
- Figure 5C needs a more in-depth analysis: if possible, I would recommend that the authors add a large figure with “KEGG metabolism category classifications” to the Discussion, adding formulas of substances to it and indicating in color which of the metabolic blocks is enhanced and which is weakened. This is really interesting for readers. In a simplified and modified form, this figure can become a graphic abstract of the article. Such an addition will significantly increase the number of citations of this article.
Response: In accordance with the reviewer suggestion, KEGG metabolism categories that are enhanced in the experimental group compared to the control group were marked in red text, while those suppressed were marked in blue text.
- Line 308-313: What determines the decrease in body weight of animals: (1) cellulose due to a decrease in the caloric content of feed (fewer calories - less weight gain)?, (2) enrichment of the bacterial flora of the intestine due to the addition of food substrates of bacteria - pectins and cellulose - to the food of dogs leads to a decrease in the consumption of calories from food by dogs?, (3) vitamins, amino acids and secondary metabolites of plants - possible analogues of animal hormones (secondary metabolites interact with receptors of dog hormones)?, (4) an increase in the immune response in the intestine to enriched microflora and due to this a switch in the dog's metabolism from the accumulation of fat to the possible fight against potentially pathogenic bacteria? or something else? In my opinion, the fourth answer option is the most correct. It is necessary to study this issue in more detail in the discussion.
Response: Thank you for your kind and insightful comments. We agree with your fourth point, which we believe is related to the F/B ratio. Accordingly, we have strengthened the discussion section to reflect this.

Reviewer 2 Report
Comments and Suggestions for Authors
The aim of the manuscript was to assess the impact of barley sprouts-supplemented diet on body weight and health markers in healthy adult beagles. The manuscript shows that dogs fed the barley sprouts diet experienced a 7.2% reduction in body weight with a significant reduction in leptin levels. There was a decrease in Firmicutes and an increase in Bacteroidetesas well as higher abundance of the Ruminococcus gnavus group. The manuscript is well written, and the data clearly presented. The experimental setup is good.
Minor comments:
Line 87: How many females and males were in each group? Was there a difference in response between female and males?
Line 129: I think IFN-r should be IFNγ.
An interesting point is that the insulin level is increased after 16 weeks for both groups. Is this an age-dependent common feature?
Figure 2: The basic IFNγ and IL-6 levels at time 0 was higher in the dogs getting BS diet. I would have expected a similar baseline.
Line 237: correct to "16S".
RDW should be spelled out first time mentioned.
In the discussion it would be preferable to spell out abbreviations such as DM, CP, DF.
Author Response
Response to Reviewers' comments
I appreciate very much for reviewer’s helpful comments on this manuscript. The manuscript has been revised according to reviewer’s suggestion thoroughly and details are explained below.
Reviewer #2:
The aim of the manuscript was to assess the impact of barley sprouts-supplemented diet on body weight and health markers in healthy adult beagles. The manuscript shows that dogs fed the barley sprouts diet experienced a 7.2% reduction in body weight with a significant reduction in leptin levels. There was a decrease in Firmicutes and an increase in Bacteroidetes as well as higher abundance of the Ruminococcus gnavus group. The manuscript is well written, and the data clearly presented. The experimental setup is good.
- Line 87: How many females and males were in each group? Was there a difference in response between female and males?
Response: Each group consisted of 4 males and 3 females and all dogs were payed or neutered. Considering the potential influence of sex differences, an analysis was conducted; however, no significant differences were observed. Thank you for your valuable feedback, and the information has been added to the Animals section of the Materials and Methods.
- Line 129: I think IFN-r should be IFNγ.
Response: The revisions were made in accordance with the reviewer comments.
- An interesting point is that the insulin level is increased after 16 weeks for both groups. Is this an age-dependent common feature?
Response: Thank you for the insightful feedback. Currently, the Dog Aging Project is being conducted on an international scale, and our research team is also conducting a project investigating the correlation between the microbiome and metabolome in relation to aging. Similar to humans, the correlation between age and blood insulin levels in dogs can be influenced by various factors such as insulin sensitivity, changes in body fat, and metabolic rate. However, the ages of the dogs used in this study are classified within the adult category (puppy < 1 year, 1 < adolescent < 3 years, 3 < young adult < 7 years, 7 < mature adult < 11 years, 11 years < senior) [1], and the absolute insulin concentrations during the experimental period (0 week versus 16 weeks) showed no statistically significant differences.
Reference
[1] Creevy, Kate E., et al. "2019 AAHA canine life stage guidelines." Journal of the American Animal Hospital Association 55.6 (2019): 267-290.
- Figure 2: The basic IFNγ and IL-6 levels at time 0 was higher in the dogs getting BS diet. I would have expected a similar baseline.
Response: Although a trend was apparent, it was not statistically significant.
- Line 237: correct to "16S".
Response: The manuscript has been revised to reflect the reviewer suggestions.
- RDW should be spelled out first time mentioned. In the discussion it would be preferable to spell out abbreviations such as DM, CP, DF.
Response: Abbreviations in the Discussion section have been revised to include the full term when mentioned for the first time.

Reviewer 3 Report
Comments and Suggestions for Authors
microorganisms-3376107-peer-review-v1
This is a very interesting academic exercise, however, what is the real application of this? Are dogs will eat this barley sprouts? Dogs are carnivores and do not see how this can have a real application.
Abstract needs to be enriched with some specific results, obtained in current study.
Paper maybe will be more appropriate for the veterinary journal.
Will be more appropriate if authors will work with RNA and not DNA, since analyzing DNA there is a possible that DNA from dead cells can be recorded and analyzed in the experimental procedure, that will give maybe misinterpretation of the obtained results.
Providing died rich on carbohydrates will lead to different problems in dogs. As I have mentioned, dogs are carnivores, and such as died based on rice and barley may have negative balance on the animals health. Authors mentioned that dogs have eat all provided food, but if they are in captivity are they have option? Eat or day. The experiment was not well set and not well planned. Authors will need to have more appropriate diet for the dogs, and then add the barley only as supplement. And then analyze the consequences of the supplementation. Moreover, the presented research plan raising question for misstarted animals.
Results are presented in more or less acceptable way. In fact, authors used the way of generating much as possible results and visual material from much of less performed experimental work. However, as I have mentioned working with DNA have specific limitations. Moreover, analysis don’t to species levels will be more appropriate to be reported. And further what are the role of that bacteria on the health performance of the dogs?
Discussion is very superficial and not really providing solid arguments regarding observed results and how in fact this can be applied.
Maybe authors can look for help from more experience colleges that can help them to use already generated data and interpreted in better way. Paper can be safe but will be more appropriate to be submitted to the veterinary journal and appropriate discussion needs to be provided for the project.
Author Response
Response to Reviewers' comments
I appreciate very much for reviewer’s helpful comments on this manuscript. The manuscript has been revised according to reviewer’s suggestion thoroughly and details are explained below.
Reviewer #3:
This is a very interesting academic exercise, however, what is the real application of this? Are dogs will eat this barley sprouts? Dogs are carnivores and do not see how this can have a real application.
Abstract needs to be enriched with some specific results, obtained in current study.
Paper maybe will be more appropriate for the veterinary journal.
Will be more appropriate if authors will work with RNA and not DNA, since analyzing DNA there is a possible that DNA from dead cells can be recorded and analyzed in the experimental procedure, that will give maybe misinterpretation of the obtained results.
Providing diet rich on carbohydrates will lead to different problems in dogs. As I have mentioned, dogs are carnivores, and such as diet based on rice and barley may have negative balance on the animals health. Authors mentioned that dogs have eat all provided food, but if they are in captivity are they have option? Eat or day. The experiment was not well set and not well planned. Authors will need to have more appropriate diet for the dogs, and then add the barley only as supplement. And then analyze the consequences of the supplementation. Moreover, the presented research plan raising question for misstarted animals.
Results are presented in more or less acceptable way. In fact, authors used the way of generating much as possible results and visual material from much of less performed experimental work. However, as I have mentioned working with DNA have specific limitations. Moreover, analysis don’t to species levels will be more appropriate to be reported. And further what are the role of that bacteria on the health performance of the dogs?
Discussion is very superficial and not really providing solid arguments regarding observed results and how in fact this can be applied.
Maybe authors can look for help from more experience colleges that can help them to use already generated data and interpreted in better way. Paper can be safe but will be more appropriate to be submitted to the veterinary journal and appropriate discussion needs to be provided for the project.
Responses:
Thank you for taking the time to review our manuscript. Your insightful comments and constructive feedback have been invaluable in further advancing our research.
As dogs evolved and became domesticated, the dogs became omnivores, not obligate carnivores, but selective carnivores. In addition, many studies have already proven that carbohydrates are an important nutrient for energy supply in dogs. The carbohydrate content in commercially available dog foods is typically 30% to 60%, with rice and barley being widely used as primary carbohydrate sources [1,2]. These grains are efficiently utilized by dogs and are recognized as beneficial ingredients for their health [1-10]. While carbohydrate content is not currently a regulatory labeling requirement for dog food, it is important to recognize that the Association of American Feed Control Officials (AAFCO) has recently initiated efforts to include carbohydrate content in the "Pet Nutrition Facts Box" [11]. The experimental diets used in this study were carefully designed to meet all nutritional requirements based on AAFCO standards. These diets were formulated by applying the nutrient requirements established by AAFCO, NRC, and FEDIAF, which are internationally recognized standards, and applied accordingly to commercially available dog foods. Accordingly, our study utilized nutritionally balanced complete diets formulated according to AAFCO standards. Individual metabolic energy requirements were meticulously calculated to provide tailored feeding for each dog. Barley sprouts was included in the diet as an additive, constituting only 2.8% of the total formula.
The proposal to submit to a veterinary journal is considered a good idea. However, our research team believes that this study, which explores the relationship between microbes and obesity in dogs, will provide insights to microbiologists and therefore it would be better to publish it in the Microorganisms journal.
We also deeply resonate with your suggestion regarding the need for species-level analysis and further research into the role of gut microbiota. Our team acknowledges the necessity of conducting additional studies to enhance analytical resolution in species-level analysis. Although shotgun sequencing can be utilized for species-level research, it requires a well-established reference genome and has technical limitations, such as the potential for errors during the assembly process. Unlike humans or mice, where genome databases are well-developed, the genomic resources for dogs remain insufficient. Therefore, 16S rRNA analysis was employed in this study, as it is currently the most appropriate method under current research conditions. This method used in this study was the most commonly used method worldwide to analyze the gut or fecal microbiome. Nevertheless, we fully agree with your perspective on the importance of species-level analysis and will incorporate this into our future research directions.
It has been proven that intestinal microbes are directly related to the health of the host. It has been scientifically proven that microbes affect not only various metabolic physiology but also mental health. Our study has newly observed the changes in intestinal microbes related to obesity as one of the causes of obesity control mechanisms in dogs through feeding barley sprouts. In line with the reviewer insightful comments, we have supplemented the results of this study in the conclusion section in more detail and even suggested ways to utilize them in the actual pet food industry.
This study aims to investigate the potential of functional ingredients and analyze changes in gut microbiota composition, ultimately contributing to the improvement of quality of life for obese dogs, addressing a growing concern in dog health. The scope of Microorganisms, which targets a broad readership interested in microbiology and metabolism-related research, is consistent with the objectives of this study. We believe that our findings can make meaningful contributions to the companion animal research community as well as to the journal’s audience.
Finally, we sincerely thank you for your outstanding expertise and valuable feedback, which have greatly enriched our work. We hope that the accumulation of research findings in the field of companion animals will continue to advance the welfare of all animals.
Reference
[1] Murray, S. M., Fahey Jr, G. C., Merchen, N. R., Sunvold, G. D., & Reinhart, G. A. (1999). Evaluation of selected high-starch flours as ingredients in canine diets. Journal of animal science, 77(8), 2180-2186.
[2] Cho, H. W., Seo, K., Lee, M. Y., Lee, S. Y., So, K. M., Kim, K. H., & Chun, J. L. (2024). Nutritional value of common carbohydrate sources used in pet foods. Journal of Animal Science and Technology, 66(6), 1282.
[3] Kore, K. B., Pattanaik, A. K., Das, A., & Sharma, K. (2009). Evaluation of alternative cereal sources in dog diets: effect on nutrient utilisation and hindgut fermentation characteristics. Journal of the Science of Food and Agriculture, 89(13), 2174-2180.
[4] Twomey, L. N., Pethick, D. W., Pluske, J. R., Rowe, J. B., Choct, M., Brown, W., & Laviste, M. C. (2002). The use of sorghum and corn as alternatives to rice in dog foods. The Journal of nutrition, 132(6), 1704S-1705S.
[5] Kempe, R., Saastamoinen, M., Hyyppä, S., & Smeds, K. (2004). Composition, digestibility and nutritive value of cereals for dogs.
[6] Axelsson, E., Ratnakumar, A., Arendt, M. L., Maqbool, K., Webster, M. T., Perloski, M., ... & Lindblad-Toh, K. (2013). The genomic signature of dog domestication reveals adaptation to a starch-rich diet. Nature, 495(7441), 360-364.
[7] De Godoy, M. R., Kerr, K. R., & Fahey Jr, G. C. (2013). Alternative dietary fiber sources in companion animal nutrition. Nutrients, 5(8), 3099-3117.
[8] Kayser, E., Finet, S. E., & de Godoy, M. R. (2024). The role of carbohydrates in canine and feline nutrition. Animal Frontiers, 14(3), 28-37.
[9] https://www.science.org/content/article/diet-shaped-dog-domestication
[10] FEDIAF, Scientific Advisory Board Carbohydrate Expert Review. 2019
[11] https://www.aafco.org/news/aafco-membership-approves-new-model-pet-food-and-specialty-pet-food-regulations/

Reviewer 4 Report
Comments and Suggestions for Authors
General comments
1. M&M lacks details.
2. The discussion and conclusion section still need to be further supplemented and improved.
3. Suggestions for further improvement of the language throughout the entire text.

Suggestions for further improvement of the language throughout the entire text.
Author Response
Response to Reviewers' comments
I appreciate very much for reviewer’s helpful comments on this manuscript. The manuscript has been revised according to reviewer’s suggestion thoroughly and details are explained below.
Reviewer #4:
This is an interesting work, but there are still the following issues.
General comments
- M&M lacks details.
- The discussion and conclusion section still need to be further supplemented and improved.
- Suggestions for further improvement of the language throughout the entire text.
Response: The manuscript has been edited by “Editage” of English editing service. The manuscript has been revised according to the suggestion of English editing service.
Specific comments
M&M
- Line86-89. Previous research has shown that gender has a certain impact on animal weight gain and other phenotypes. What is the reason why gender factors were not considered in this study? In addition, are the selected experimental animals half male and half female?
Response: Each group consisted of 4 males and 3 females and all dogs were payed or neutered. Considering the potential influence of sex differences, an analysis was conducted; however, no significant differences were observed. Thank you for your valuable feedback, and the information has been added to the Animals section of the Materials and Methods.
- Line 95-96. There is a lack of description in the experimental design regarding animal management methods, such as whether they are free to feed and the daily feeding time. In addition, this experimental design indicates that all animals have no leftover feed. How to determine the feeding amount?
Response: All experimental methods were conducted in accordance with AAFCO (Association of American Feed Control Officials) standards, and this information has been added to the Animals section of the Materials and Methods.
- Line 90. “2.8% barley sprouts” What is the basis for setting the amount of addition?
Response: In accordance with a previous study conducted on mice, sprouted barley was administered proportionally to body weight.
- Line 135. How to collect fecal samples?
Response: Fecal samples were collected from naturally excreted feces, and this information has been added to the Fecal sample collection and microbiota analysis section of the Materials and Methods.
- Statistical analysis This part is missing.
Response: The 2.5 Statistical analysis section has been added to the Materials and methods section.
Result
- Line 158. The overall feed intake of the animals was not provided.
Response: In accordance with the comments, the relevant content has been added to the Result section.
- Line 210. How to determine the digestibility of ME?
Response: The ME equation has been added to the legend of Table 2.
- Line 279. The genus level profile of gut microbiota is not shown. In addition, there is a lack of screening criteria, such as the relative abundance of the displayed bacterial genera being all greater than 1%.
Response: Interestingly, differences were observed at the phylum level, but at the genus level, only the Ruminococcus gnavus group showed differences. To verify this result, we applied various analytical methods, and the widely used LEfSe analysis for identifying gut microbiota markers also confirmed the same result. Thank you for your valuable suggestion, and we have added this information to the Results section.
Discussion
- Abbreviations that first appear need to indicate their full name.
Response: Abbreviations in the Discussion section have been revised to include the full term when mentioned for the first time.
Conclusion
- Line 371. Lack of summary and condensation of research results and highlights, and the results of this study can only prove that the experimental treatment has a certain impact on body weight and gut microbiota, and cannot explain that body weight is regulated through gut microbiota.
Response: Upon reviewer comment the Conclusion section in the manuscript has been revised and re-wirted.

Round 2
Reviewer 1 Report
Comments and Suggestions for Authors
Most of the shortcomings have been corrected, but the data in the following articles in the figures should be presented in the form of a box analysis. This will increase the readers' confidence in the research results.
Reviewer 3 Report
Comments and Suggestions for Authors
microorganisms-3376107-peer-review-v2
In previous review stage of the manuscript, I have stated different concern about the manuscript, however, in provided revised version only “cosmetic” adjustments in the manuscript were provided, however, most of the questions and concerns were ignored.
This is a very interesting academic exercise, however, what is the real application of this? Are dogs will eat this barley sprouts? Dogs are carnivores and do not see how this can have a real application.
Abstract needs to be enriched with some specific results, obtained in current study.
Paper maybe will be more appropriate for the veterinary journal.
Will be more appropriate if authors will work with RNA and not DNA, since analyzing DNA there is a possible that DNA from dead cells can be recorded and analyzed in the experimental procedure, that will give maybe misinterpretation of the obtained results.
Providing died rich on carbohydrates will lead to different problems in dogs. As I have mentioned, dogs are carnivores, and such as died based on rice and barley may have negative balance on the animals health. Authors mentioned that dogs have eat all provided food, but if they are in captivity are they have option? Eat or day. The experiment was not well set and not well planned. Authors will need to have more appropriate diet for the dogs, and then add the barley only as supplement. And then analyze the consequences of the supplementation. Moreover, the presented research plan raising question for misstarted animals.
Results are presented in more or less acceptable way. In fact, authors used the way of generating much as possible results and visual material from much of less performed experimental work. However, as I have mentioned working with DNA have specific limitations. Moreover, analysis don’t to species levels will be more appropriate to be reported. And further what are the role of that bacteria on the health performance of the dogs?
Discussion is very superficial and not really providing solid arguments regarding observed results and how in fact this can be applied.
Maybe authors can look for help from more experience colleges that can help them to use already generated data and interpreted in better way. Paper can be safe but will be more appropriate to be submitted to the veterinary journal and appropriate discussion needs to be provided for the project.